# Modeling the Impact of Meniscal Tears on von Mises Stress of Knee Cartilage Tissue

**DOI:** 10.3390/bioengineering10030314

**Published:** 2023-03-01

**Authors:** Oleg Ardatov, Viktorija Aleksiuk, Algirdas Maknickas, Rimantas Stonkus, Ilona Uzieliene, Raminta Vaiciuleviciute, Jolita Pachaleva, Giedrius Kvederas, Eiva Bernotiene

**Affiliations:** 1Faculty of Mechanics, Vilnius Gediminas Technical University, LT-10223 Vilnius, Lithuania; 2Department of Regenerative Medicine, State Research Institute Centre for Innovative Medicine, LT-08410 Vilnius, Lithuania; 3Faculty of Medicine, Vilnius University, LT-03101 Vilnius, Lithuania; 4Faculty of Fundamental Sciences, Vilnius Gediminas Technical University, LT-10221 Vilnius, Lithuania

**Keywords:** cartilage, femur, finite element method, hyperelasticity, meniscus, tibia

## Abstract

The present study aims to explore the stressed state of cartilage using various meniscal tear models. To perform this research, the anatomical model of the knee joint was developed and the nonlinear mechanical properties of the cartilage and meniscus were verified. The stress–strain curve of the meniscus was obtained by testing fresh tissue specimens of the human meniscus using a compression machine. The results showed that the more deteriorated meniscus had greater stiffness, but its integrity had the greatest impact on the growth of cartilage stresses. To confirm this, cases of radial, longitudinal, and complex tears were examined. The methodology and results of the study can assist in medical diagnostics for meniscus treatment and replacement.

## 1. Introduction

Meniscal tears in knee joints are prevalent, with an average annual incidence of 60–66 cases per 100,000 population [1,2]. The tears can be classified according to geometric orientation [3]. Longitudinal tears occur between circumferential collagen fibers, which are unstable and cause mechanical symptoms or the partial fixation of the knee. Radial tears destroy circumferential collagen fibers and affect the ability of the meniscus to absorb tibiofemoral loads [4]. Complex or degenerative tears comprise two or more tear patterns. They are more common in elderly people and are associated with osteoarthritis (OA) development in the knee [5]. It is known that the relationship between OA and meniscal tears is complex. Degenerative meniscal tear and meniscal extrusion often lead to decreased meniscal coverage of cartilage, impaired mobility, and the initiation or progression of OA [6,7], and the development of OA may accelerate the damage of the meniscus [8]. From a medical point of view, knee menisci in OA undergo morphological and molecular changes, such as a reduction in the cell population, the disorganization of the extracellular matrix, and disturbances in protein synthesis and expression [9]. In human OA menisci, there is a significant decrease in cells compared with cells in non-OA menisci [10]. A low number of cells reduces the synthesis of collagens, glycosaminoglycans, and other proteins, resulting in clefts in the extracellular matrix [11]. The expression of type VI collagen in the articular cartilage of patients with OA is higher than in healthy ones [12]. Furthermore, OA is associated with progressive joint pain and the restriction of movements, which can have a large negative impact on life quality. OA is also associated with many comorbidities, such as diabetes and cardiovascular diseases [13]. OA treatment therapy mostly focuses on symptomatic treatment, including the management of pain, inflammation, and degeneration, but the regeneration of degenerated cartilage has not been achieved [14].

The in silico modeling of meniscal tears may help clarify the biomechanical processes that transpire in the case of a damaged meniscus. The advantages of this approach include the avoidance of surgical intervention, the possibility of evaluating different scenarios of disease progression, as well as the possibility of the parameterization of models, which, in turn, allows for the simulation of different tissue states and the prediction of possible outcomes under certain load conditions. The disadvantages of the in silico method include the loss of accuracy when extracting geometry from medical images; the files require additional processing in which the shape of numerical models is inevitably distorted. In addition, modeling the mechanical behavior of biological tissues requires the integration of a more complex mathematical apparatus. This is especially important for soft tissues, such as menisci or cartilage, because their response to external load is often nonlinear. Additional difficulties may be caused by a lack of information on the mechanical properties of biological tissues. Several tissues are insufficiently studied, and the values of elastic constants and other values may vary depending on various factors, such as age, sex, diseases, and others.

In previously described numerical models of knee joint components [15], bones, menisci, and cartilage are considered perfectly elastic isotropic materials, although many biological tissues demonstrate nonlinear and anisotropic behavior [16]. Thus, the ability to predict the behavior of real knee components remained limited. Later studies described more complex material models [15], where meniscus material is represented by the Neo–Hookean hyperelastic material model [17]. This material model was determined to extend linear elasticity to nonlinear regions of large strain, which is more realistic than linear elasticity. However, in that model, both bones and cartilage are there represented as a rigid continuum and therefore fail to represent the natural characteristics of cartilage tissue. The model developed for the Open Knee project [18] uses a set of similar material models, but the cartilage is modeled using the Neo–Hookean material instead of being rigid, whereas the menisci are modeled using the Fung orthotropic material. These changes provide a slight improvement because the cartilage deformability and anisotropy of the meniscus are considered. However, the previously mentioned limitations of the Neo–Hookean materials continue to hinder this model.

Brown et al. [19] studied several hyperelastic models, such as Arruda–Boyce, Mooney–Rivlin, Neo–Hookean, Ogden, Polynomial, and Yeoh, and showed that the Mooney–Rivlin model is best suited for the normalized response to the tension and compression of cartilage specimens. These experiments were carried out on 57 bovine patella articular cartilage specimens; both healthy and osteoarthritis specimens were tested at strain rates of 0.1 and 0.025 s^−1^. Various hyperelastic stress–strain curves were applied to the experimental data—Arruda–Boyce, Mooney–Rivlin, Neo–Hookean, Ogden, Polynomial, and Yeoh. The Mooney–Rivlin and Yeoh models were found to fit best, with r^2^ values of 0.999. In the Neo–Hookean model, r^2^ was 0.998.

Some studies [20,21,22,23] presented numerical models of the knee joint to study the impact of meniscus tears on the mechanical responses of cartilage. Almost all of these studies have declared that changes were caused by increasing the contact stress on the contact area of knee joint tissues.

However, these studies were based on the assumption that the bone tissue is a completely rigid body. This approach may be close to reality because of the relatively higher stiffness of the bone compared with the cartilage or meniscus, but the deformation of the tibia and femur may affect the distribution of forces on the contact area. This was validated by Li et al. [24], who examined the effects of shear stresses on the cartilage and menisci in the case of standing. The three-dimensional numerical model was evaluated using the finite element method.

The present study aims to examine the changes in the stressed state of cartilage caused by meniscal tears, with verification of bone elasticity and hyperplastic properties of cartilage. We assume that the analysis of stress values, as well as their distribution over the contact surface, and the presence of stress concentrators, can serve as a criterion for evaluating the functionality of the meniscus. To confirm this, we examined several cases of meniscus damage, and we obtained the mechanical properties of the meniscus that are characteristic of the fourth and third stages of OA and assigned them to the studied models. To obtain the most accurate description possible, we applied the nonlinear theory of elasticity and used the Mooney–Rivlin material model for the mechanical behavior of cartilage.

## 2. Materials and Methods

### 2.1. Geometry of the Model

The development of a three-dimensional numerical model (Figure 1) of the knee joint included several steps. To obtain the initial geometry of the femur and tibia, computed tomography images of a 60-year-old female diagnosed with third-stage OA were used, which were processed using the free open-source software, 3D Slicer [25]. In order to perform CT scanning, the General Electric Discovery GE CT750 HD was used. Then, the resulting geometry was further processed using the MeshLab program [26], which was performed to remove noise and smooth the surfaces. Finally, the output STL file was exported to the SolidWorks software [27] environment, where, using the ScanTo3D module, the final rendering of the mesh took place, and surfaces were transformed into solid-state models of the femur and tibia. Furthermore, in the same SolidWorks environment, cartilage congruent to articular surfaces was formed for bone components. Their thicknesses (2 mm) were assumed to be the same throughout the volume. In the next stage, menisci were added to the existing bones and cartilage. Their geometry, as well as methods of damage modeling, are described in more detail in Section 2.5 (Modeling of menisci damage).

Notably, the initial geometry went through three processing steps, during which, the unavoidable distortion of the geometric shape and volume occurred. Although the model reflects the characteristic curvature of bones, because of the numerous refinements of the initial geometry, the errors in geometry are undeterminable.

Additionally, the absence of ligaments constitutes another limitation of the model. Therefore, the model is only assessed for compression loads, because calculating cases with rotation and flexion can provide unreliable results.

### 2.2. Problem Formulation

To verify the mechanical behavior of the model, the nonlinear theory of elasticity was used [27]. In nonlinear dynamic analysis, the equilibrium equations of the dynamic system at time step *t* + ∆*t* are:(1)[M]{U″} t+Δt (i)+[C]{U′} t+Δt(i)+[K] t+Δt(i){ΔU} t+Δt(i)={R}  t+Δt−{F} t+Δt(i−1),
where [*M*] is the mass matrix of the system, [*C*] is the damping matrix of the system, *t* + ∆*t*[*K*](*i*) denotes the stiffness matrix of the system, *t* + ∆*t*{*R*} denotes the vector of externally applied nodal loads, *t* + ∆*t*{*F*}(*i* − 1) denotes the vector of internally generated nodal forces at iteration (*i* − 1), *t* + ∆*t*[∆*U*](*i*) denotes the vector of incremental nodal displacements at iteration (*i*), *t* + ∆*t*{*U*’}(*i*) denotes the vector of total velocities at iteration (*i*), and [*M*]*t* + ∆*t*{*U*’’}(*i*) represents the vector of total accelerations at iteration (*i*), where damping matrix [*C*] was neglected, [*C*] = 0.

Using the implicit time integration Newmark–Beta scheme and employing Newton’s iterative method, the above equations are expressed in the following form:(2)[K] t+Δt(i)  {ΔU}(i)={R} t+Δt(i),
where *t* + ∆*t*{*R*}(*i*) denotes the effective load vector and *t* + ∆*t*[*K*](*i*) represents the effective stiffness matrix. The three-dimensional nonlinear analysis was performed using SolidWorks software.

### 2.3. Mechanical Properties of Model Components

The bone tissue is assumed to be perfectly elastic. The Young’s modulus for both the femur and the tibia was set to 7.3 GPa, the Poisson’s ratio was set to 0.3 [24], and the cartilage is modeled as a hyperelastic (Mooney–Rivlin) continuum.

The Mooney–Rivlin strain energy density function is expressed as a two-constant formulation [28]:(3)$$w=C_{1}\left(I_{1}-3\right)+C_{2}\left(I_{2}-3\right)+\frac{1}{2}K{\left(I_{3}-1\right)}^{2},$$
where *C*_1_ and *C*_2_ are the first and second material constants, respectively, related to the distortional response, and *K* is the material constant related to the volumetric response. *I*_1_, *I*_2_, and *I*_3_ are the reduced invariants of the Cauchy–Green deformation tensor and can be expressed in terms of principal stretch ratios.

The material constant *K* can be determined as:(4)$$K=\frac{6\left(C_{01}+C_{10}\right)}{3\left(1-2\upsilon \right)};$$
where *υ* is Poisson’s ratio. In our research, it is set to 0.4995. The elastic constants *C*_1_ and *C*_2_ for tibia cartilage and femur cartilage are set according to the range reported by [29]. They are presented in Table 1.

The mechanical properties of the meniscus were determined experimentally by compressing the specimen using a single-column force tester. The methodology of the physical experiment is described in the next section.

### 2.4. Experimental Determination of Meniscus Mechanical Properties

Owing to a lack of data for the elasticity constants of the meniscus, the mechanical properties were determined experimentally using the computer-controlled tension–compression test system, Mecmesin MultiTest 2.5-i (Mecmesin Limited, Horsham, UK, maximum load: 2500 N; maximum sample diameter: 134 mm; load sensor measurement error: ± 0.1%; speed range: 1–1000 mm/min). The testing machine was controlled using the Emperor software (Mecmesin Ltd., Horsham, UK) (Figure 2a).

The cylindrical specimens of the human meniscus (6 mm diameter, 6 mm length, Figure 2b) were obtained from two different patients. Patient cartilage samples, following informed consent, were obtained during joint replacement surgery from Vilnius Santaros Clinics, according to the Bioethical committee permission No 158200-14-741-257. Clinical information: the first meniscus was obtained from a knee joint with stage IV OA and the second one from a knee with stage III OA. Both patients were women. Additional data are presented in Table 2.

Testing conditions: an axial load with a constant speed of 8 mm/min was used for the compression test, and 14 specimens were compressed (7 from each patient). The stress–strain curves represent the average values of tested specimens, shown in Figure 3.

From Figure 3, the meniscus stress–strain curve of the first patient shows greater stiffness. The response may be explained by the clinical state of the studied menisci. To clarify the difference in tissue structure, a histological evaluation of the menisci specimens was performed (see Figure 4).

The structures of both menisci are shown in Figure 4a,b—the surface of the first meniscus is wavy, fibrillating, and the extracellular matrix has many clefts, whereas the meniscus of the second patient has fewer clefts and its surface is smoother. Upon greater magnification (20×), the distribution of cells (chondrocytes) can be observed, as shown in Figure 4c,d. The first meniscus (Figure 4c) consists of just a few haphazardly arranged cells, whereas the second meniscus (Figure 4d) contains more cells in the superficial layer. On the greatest magnification (40×), the morphology of the cells can be observed. The cells of the first meniscus (Figure 4e) are round, with pale cytoplasm and without nuclei (apoptotic cells) in the superficial layer, which indicates that those cells are dead. The cells of the second meniscus (Figure 4f) are arranged in rows and have elongated, hyperchromatic nuclei, which implies that they are active and can divide. These observations reveal that the first meniscus is more deteriorated than the second one.

### 2.5. Modeling of Meniscus Damage

The damage to the meniscus is modeled by removing volume from its initial geometry. The decrease in meniscus mass can be caused by trauma, fractures, or degenerative processes. The meniscus loses parts of its initial geometry, so it cannot transfer loads effectively. Possible variants of damage [30] are presented in Figure 5.

In Figure 5, to simulate the longitudinal tear of the meniscus, a strip of material was removed from its initial volume. Specifically, a longitudinal thorough-hole was obtained, and the meniscus, in turn, lost 17% of the initial mass. The immediate area of contact between the cartilage surfaces has not changed. The cartilage still interacts through the meniscus; however, the resistance of the meniscus to external loads should decrease.

With a radial tear, there are no through-holes in the damaged meniscus, and unlike a longitudinal rupture, with a radial meniscus, it only lost 8% of its initial mass. However, because of the removal of material, the direct contact area between the cartilage surfaces increased, and since the stiffness of the cartilage is higher than that of the menisci, this should cause more intense mechanical interaction between cartilage surfaces.

With a complex tear, the meniscus has lost about half of its initial mass. In addition, the contact area of the cartilage interaction has increased, which should affect the stressed state.

### 2.6. Calculation Cases, Boundary Conditions and Mesh

To verify the effect of meniscus damage on the stressed state of cartilage, numerical calculations for both patients with different types of meniscus damage and their combinations were performed. The schematization of the load is shown in Figure 6a. The bottom surface of the tibia is constrained from any motion while the vertical compression load (500 N) is supplied to the upper surface of the femur. To solve the equilibrium equations, the Intel Direct Sparse solver was used. To effectively adapt the finite element mesh to the complex curvature of the model, meshes were applied with tetrahedral finite elements (Figure 6b). Calculation cases and mesh data are presented in Table 3.

## 3. Results and Discussion

In the course of the study, stressed state plots were obtained for the meniscus of two different patients with different damage types: undamaged meniscus, meniscus with a longitudinal tear, meniscus with a radial tear, and meniscus with a complex tear. The plots are shown in Figure 7. With an undamaged meniscus (Figure 7a,e), the stresses are distributed almost uniformly; no stress concentrators are found. This observation is in good agreement with the results reported previously [15,28]. The maximum values of stress (see Figure 8) are also small (1.31 MPa for the first patient and 1.24 MPa for the second patient). A comparison of the obtained values with the ultimate strength of cartilage suggests that the strength capacity is sufficient. The range of ultimate strength for cartilage from different sources is 25–40 MPa [15,16,17].

With a longitudinal tear of the meniscus (Figure 7b,f), stress concentrators appear in two regions. In addition, the stress value increased by 63% for the first patient and 53% for the second patient. Notably, the areas of stress concentrators with maximum stress values are larger in the second patient. This may be because the meniscus of the second patient is less deteriorated and its load-transferring ability is more effective. That also explains why the average values of the first patient’s meniscus stresses are higher than the second one.

With a radial tear, an increase in the stress value (Figure 7c,g) is also observed. Compared with an undamaged model, the average stresses increase by 81% for the first patient and 86% for the second patient. However, the area of sites with stress concentrators is reduced.

With a complex tear of the meniscus (Figure 7d,h), both the maximum stress values and the area of the stress concentrators increase. Compared with the undamaged meniscus, the stresses increase more than three times. The increased area of stress concentrators also indicates that the meniscus can no longer perform its function of mitigating and transferring the load.

The stress distribution on tibia cartilage is shown in Figure 9. The tendency for stress to increase remains the same. In the case of an undamaged meniscus, no stress concentrators are observed (Figure 9a,e). In the case of a longitudinal tear, there are two areas with stress concentrators observed (Figure 9b,f), whereas only one area is observed in the case of a radial tear (Figure 9c,g). We can hardly treat this as a positive factor, however, because the values of stress in the case of the radial tear are higher. This phenomenon can be explained by distortion between contacting surfaces, which occurs because of the radial tear and because the meniscus does not transfer loads effectively.

In the case of complex tears, four areas of stress concentrators are observed (Figure 9d,h). The maximum stress values (see Figure 10) in the first and second patients are almost the same. Furthermore, the meniscus is no longer functional (4.21 MPa in the first patient and 4.19 MPa in the second patient), so medical treatment and meniscus replacement are required.

The areas of stress concentrators for Patient No. 2 exceeded the stress values for Patient No. 1; however, from a medical point of view, the meniscus of Patient No. 2 was less deteriorated. This reveals that the meniscus of Patient No. 2, transferred the loads more smoothly, and the increased contact area affected the maximum values of stresses. In addition, the distribution of stress was more uniform for Patient No. 2, whereas pronounced peaks of high stress were observed in Patient No. 1. From a mechanical point of view, the peaks are associated with the appearance of microcracks.

Based on the results of this study, complex tears are the most dangerous, but in cases of longitudinal and radial tears, stresses are distributed more or less the same. In the case of the undamaged meniscus, the stress distribution on the cartilage remains approximately the same, regardless of the properties of the meniscus, but the values of stresses can be varied according to the mechanical properties of the meniscus. Although the level of meniscus deterioration was different between the two patients, our results revealed that it is essential to evaluate the general condition of the joint.

Notably, biomechanical changes in the meniscus are associated with the progress of OA [31]. Meniscus injury leads to narrowed joint space, altered joint stability, deteriorated transmission of the load, cartilage degeneration, pain, and impaired function [32]. Meniscal repair or replacement can be applied to prevent the onset or progression of OA, but there is no effective way to restore the damaged cartilage [6]. One major drawback is the lack of efficient models that can predict the outcomes of novel therapies. Thus, the proposed model may assist in the development of novel intraarticular treatment or meniscus replacement strategies.

## 4. Conclusions

We developed a numerical three-dimensional model of the knee joint that reflects its anatomical geometry and can assess the stress state of its components. To obtain a more detailed picture of the mechanical behavior of the knee joint components, we accounted for the nonlinear properties of both the cartilage and meniscus. To simulate the behavior of cartilage tissue, the Mooney–Rivlin model was used, whereas the menisci were assigned a stress–strain curve, obtained experimentally from samples of two different patients. The following statements summarize the major contributions of this work:The distribution of stresses on the cartilage surfaces varied slightly depending on the mechanical properties of the undamaged meniscus. The differences in stress distribution were more noticeable in cases of different damage types. Thus, the geometric shape and mechanical condition of the knee joint have the greatest influences on stress distribution;The major factors, such as stress values, their distribution on the contact surfaces, as well as the clinical condition of the meniscus and cartilage tissues, should be evaluated together, since they may be individually uninformative;The results of the study and the developed methodology can be useful for further studies of the mechanical aspects occurring in the knee joint, as well as for modeling artificial meniscus and predicting cartilage responses.

## Figures and Tables

**Figure 1 bioengineering-10-00314-f001:**
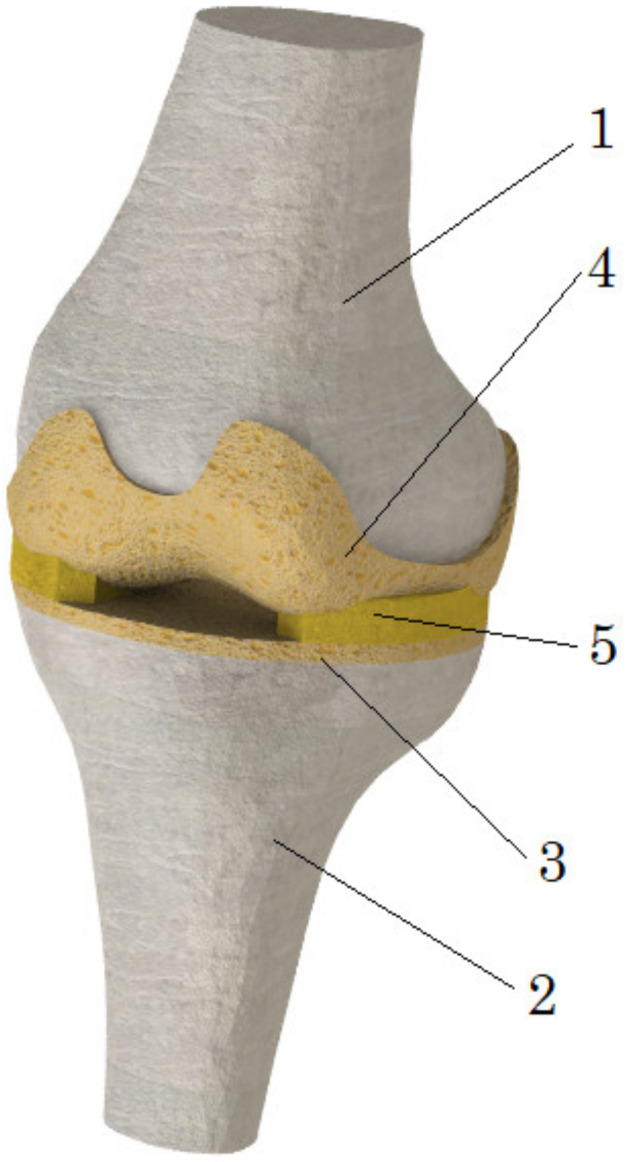
The numerical model of the knee joint: 1—femur; 2—tibia; 3, 4—cartilage; 5—meniscus.

**Figure 2 bioengineering-10-00314-f002:**
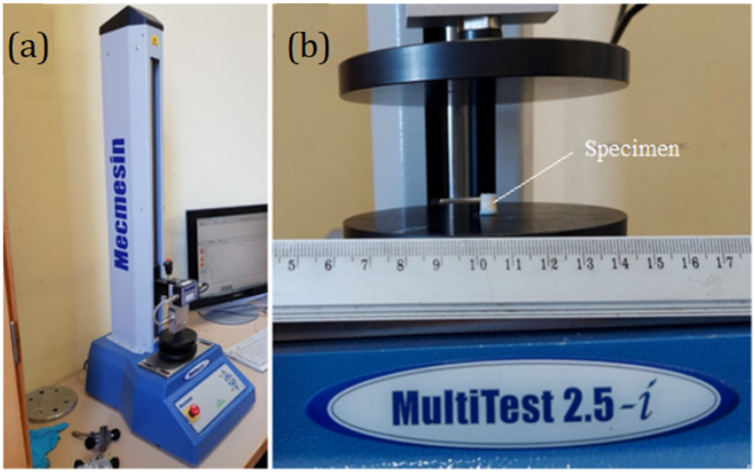
(**a**): Single-column force tester, MultiTest 2.5-i; (**b**): Cylindrical specimen extracted from the human meniscus.

**Figure 3 bioengineering-10-00314-f003:**
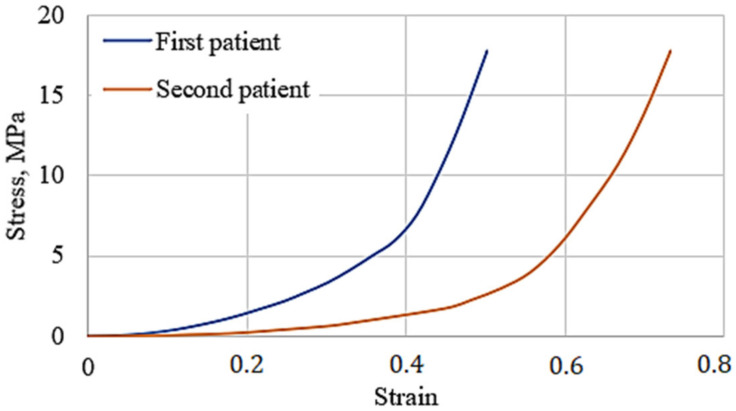
Stress–strain curves of the human meniscus.

**Figure 4 bioengineering-10-00314-f004:**
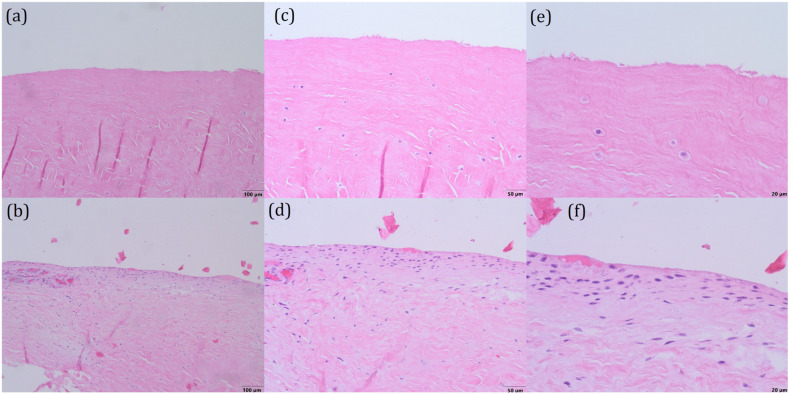
Histological and immunohistochemical evaluation of menisci stained with hematoxylin and eosin: (**a**,**b**), magnification 10×; (**c**,**d**)—magnification 20×; (**e**,**f**)—magnification 40×; (**a**,**c**) and (**e**)—Patient No. 1; (**b**,**d**,**f**)—Patient No. 2.

**Figure 5 bioengineering-10-00314-f005:**
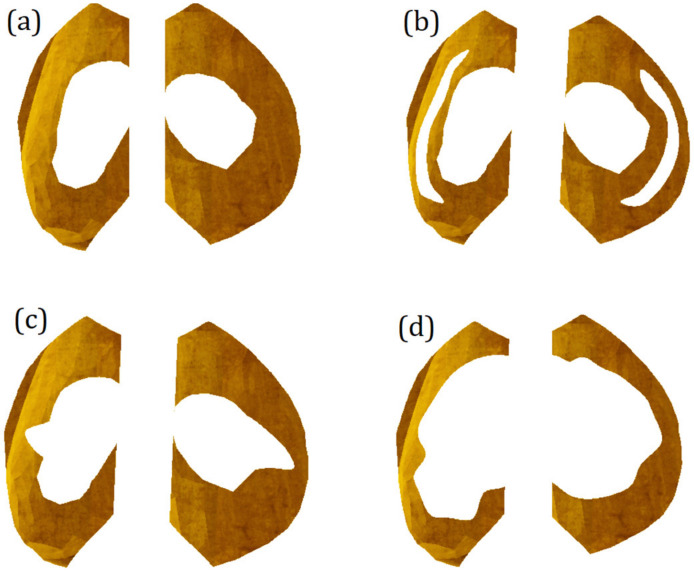
Possible variants of meniscus damage modeled by removing volume from its initial geometry: (**a**) relatively undamaged meniscus; (**b**) meniscus with a longitudinal tear; (**c**) meniscus with a radial tear; (**d**) complex tear.

**Figure 6 bioengineering-10-00314-f006:**
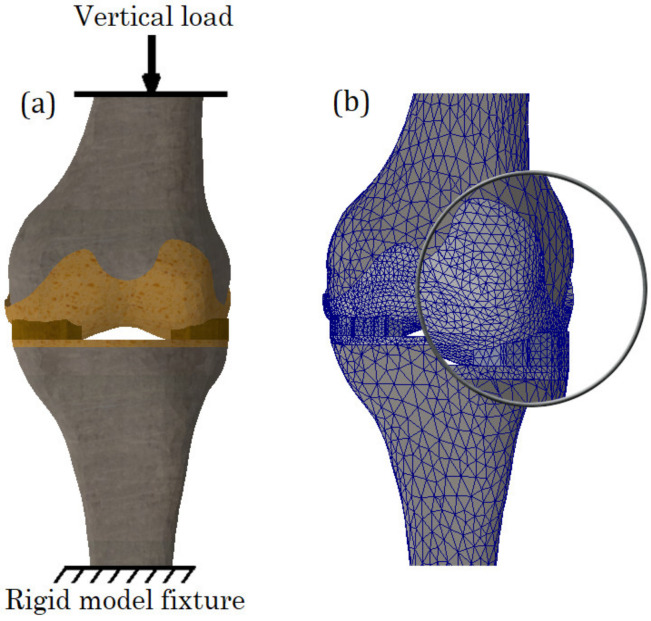
(**a**): Schematization of load; (**b**): Finite element mesh.

**Figure 7 bioengineering-10-00314-f007:**
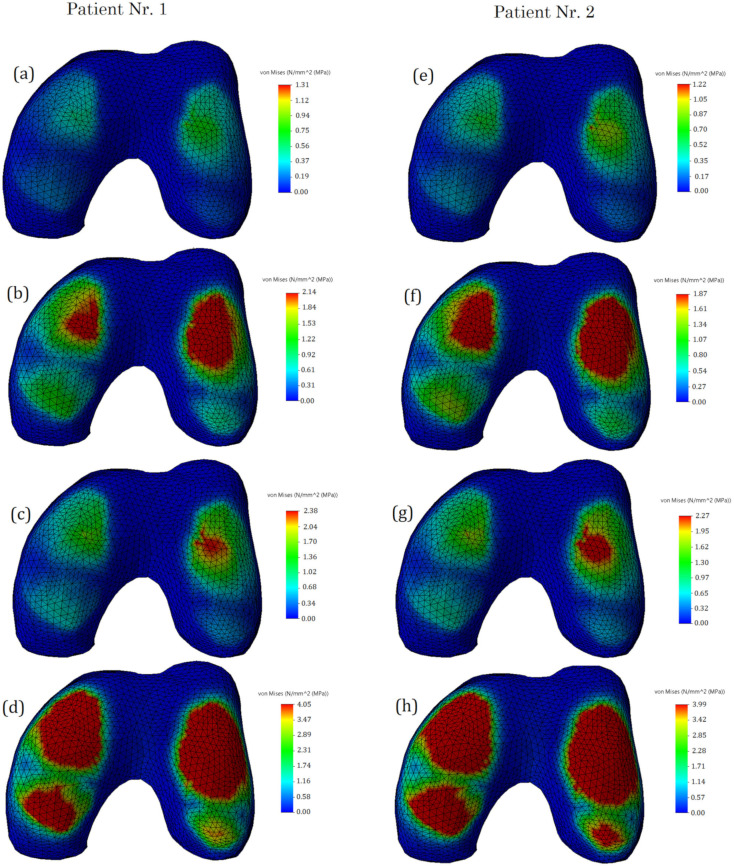
The von Mises stress on femur cartilage: (**a**,**e**)—undamaged meniscus; (**b**,**f**)—meniscus with a longitudinal tear; (**c**,**g**)—meniscus with a radial tear; (**d**,**h**)—meniscus with a complex tear.

**Figure 8 bioengineering-10-00314-f008:**
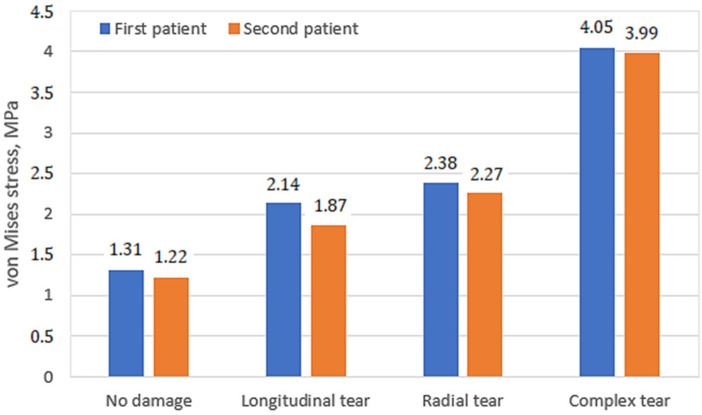
Maximum von Mises stress values on femur cartilage.

**Figure 9 bioengineering-10-00314-f009:**
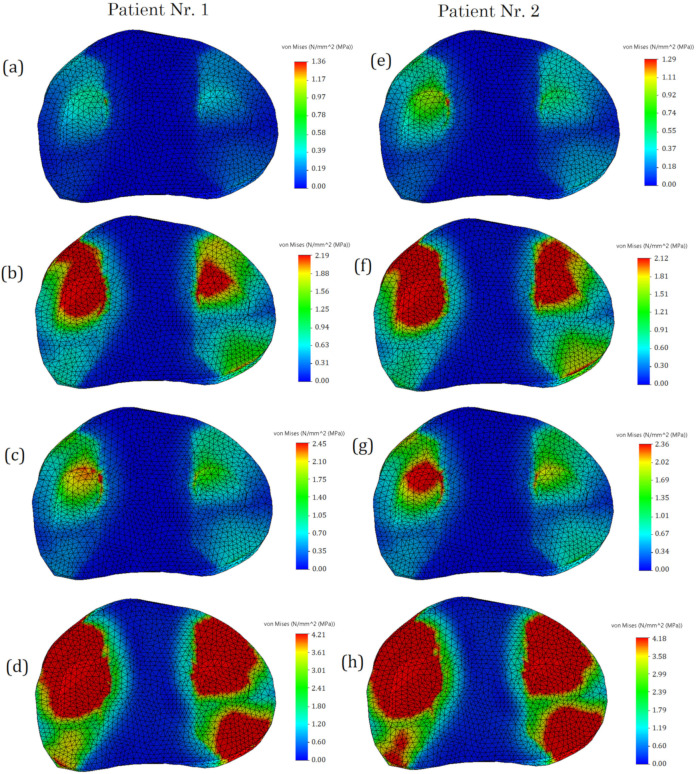
Von Mises stress on tibia cartilage: (**a**,**e**)—undamaged meniscus; (**b**,**f**)—meniscus with a longitudinal tear; (**c**,**g**)—meniscus with a radial tear; (**d**,**h**)—meniscus with a complex tear.

**Figure 10 bioengineering-10-00314-f010:**
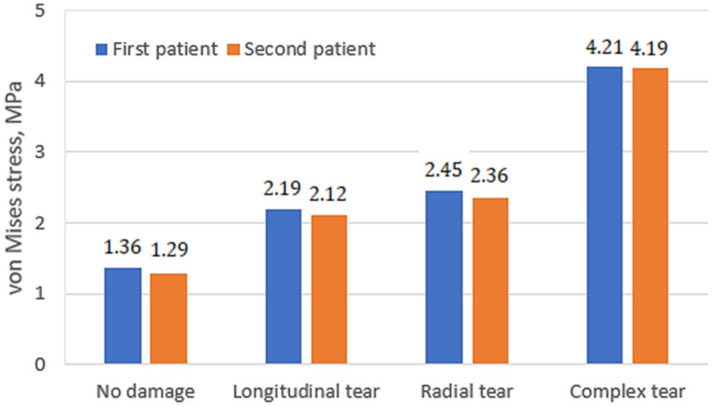
Maximum von Mises stress values on tibia cartilage.

**Table 1 bioengineering-10-00314-t001:** Elastic constants of tibia cartilage and femur cartilage.

Model Component	*C*_1_, MPa	*C*_2_, MPa
Cartilage (tibia)	1.13	0.32
Cartilage (femur)	1.08	0.63

**Table 2 bioengineering-10-00314-t002:** General clinical data of menisci donors.

Patient No.	Sex	Age	Body Mass Index	Diagnosis
1	Female	61	41.0	Knee osteoarthritis stage IV
2	Female	60	34.4	Knee osteoarthritis stage III

**Table 3 bioengineering-10-00314-t003:** Mesh data on various model types.

Calculations Cases	Number of Finite Elements	Number of Nodes
No meniscus damage	53,443	89,871
Longitudinal meniscal tear	51,252	87,799
Radial meniscal tear	52,884	88,345
Complex meniscal tear	50,986	87,658

## Data Availability

The data presented in this study are available on request from the corresponding author.

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
