# Peer review of "Modeling the Impact of Meniscal Tears on von Mises Stress of Knee Cartilage Tissue"

_bioengineering, 2023, doi:10.3390/bioengineering10030314_

Round 1

Reviewer 1 Report

The manuscript investigated the stress state of cartilage due to various meniscal tear models. The structure of the manuscript is okay. However, there are following major issues:

1.     The English in the manuscript is not readable in many places. I suggest the authors refer to some professional English Editing serve to improve the English.

2.     The professional writing and presentation in the manuscript should be improved. For example, some key parts, including the information about the patient, the CT data, etc., are missing. Whether the consent has been obtained? And is the study protocol been approved? I suggest the authors refer to some experienced researchers to improve the writing and presentation.

3.     The number in Figure 7 is not readable and should be improved.

Author Response

Point 1: The English in the manuscript is not readable in many places. I suggest the authors refer to some professional English Editing serve to improve the English.

Response 1: Dear reviewer, thak you for reviewing our manuscript. We’ve referred to the proofreading experts for errors and grmmar mistakes and professional English language in our work is now improved.

Point 2: The professional writing and presentation in the manuscript should be improved. For example, some key parts, including the information about the patient, the CT data, etc., are missing. Whether the consent has been obtained? And is the study protocol been approved? I suggest the authors refer to some experienced researchers to improve the writing and presentation.

Response 2: Thank you for suggestions. We totally agree with your comments, and have revised the manuscript according to them: the explanation on developing the numerical model is added, the reference on used software is added too. The clinical data of menisci donors is added. The information of obtained consent is supplemented in subsection 2.4. and in acknowledgements. We’ve made the overall revise of our manuscript in order to make it in more appropriate way.

Point 3: The number in Figure 7 is not readable and should be improved.

Response 3: Thank you for the response, the readability of numbers in figures is improved now.

Reviewer 2 Report

Authors developed a numerical three-dimensional model to assess the stress state of cartilage caused by various meniscus tears. Some new findings in this study. Nevertheless, I found some points need to be discussed.

- Abstract

The abstract should be reorganized as the methodology was not clearly elaborated, and some critical and interesting results were not presented.

Introduction

·       The first paragraph of the introduction includes information not relevant for this study, such as the used of stem cells and scaffold etc.

·       It is unnecessary to introduce the composition of this study in the last paragraph of the introduction.        Instead, the authors should highlight the innovation of their research conception, provide a statement on the outcomes and the clinical significance of this study.

Methods

Geometry of the model:  In present study, the numerical model of knee joint was constituted by five parts (1. femur, 2. tibia; 3, 4 cartilages; 5. meniscus). However, ligaments around the knee (such as ACL/PCL, lateral/medial collateral ligaments, et al) are quite important to the stability of knee joint. If possible, it is better to build a model contains main ligaments.

Boundary conditions and mesh:  “The models were characterized by range of 262 353-271 826 degrees of freedom: it is meshed with tetrahedral finite elements due to its curvature (Figure 6b). The number of elements – 50 986-53 443, the number of nodes – 87 658-89 871.” It is incomprehensive what the numbers is referred to

Conclusions

Way too long.  Point 2 should be discussion materials

General comments

The manuscript was not well written, with typo and unstructured sentences.  Eg the 4th point in the conclusion.

Author Response

Dear Reviewer, thank You for revising our manuscript. All your sugestions are clear, so we’ve tried our best to follow them.

Point 1:. The abstract should be reorganized as the methodology was not clearly elaborated, and some critical and interesting results were not presented.

Response 1: We‘ve understood the point and reorganized the abstract and the most important results are now pronounced.

Point 2: Introduction.

The first paragraph of the introduction includes information not relevant for this study, such as the used of stem cells and scaffold etc.

It is unnecessary to introduce the composition of this study in the last paragraph of the introduction.  Instead, the authors should highlight the innovation of their research conception, provide a statement on the outcomes and the clinical significance of this study.

Response 2: Thank you for suggestions. The non-relevant information is now removed. The unnecessary senteces are removed. Main conception and significance of the paper is added and expressed.

Point 3:. Methods

Geometry of the model:  In present study, the numerical model of knee joint was constituted by five parts (1. femur, 2. tibia; 3, 4 cartilages; 5. meniscus). However, ligaments around the knee (such as ACL/PCL, lateral/medial collateral ligaments, et al) are quite important to the stability of knee joint. If possible, it is better to build a model contains main ligaments.

Response 3: Yes, we agree with this statement. In case of our model only normal compressive loads can be applied and verified, without ligaments we can’t reaaly examine flexion or rotary loads. We’ve included this limitation in our text.

Point 4:. Boundary conditions and mesh:  “The models were characterized by range of 262 353-271 826 degrees of freedom: it is meshed with tetrahedral finite elements due to its curvature (Figure 6b). The number of elements – 50 986-53 443, the number of nodes – 87 658-89 871.” It is incomprehensive what the numbers is referred to.

Response 4: We added explanations why these values can vary and added table in which indicated to which model of damaged meniscus these values are applyed to.

Point 5:. Conclusions. Way too long.  Point 2 should be discussion materials

Response 5: Thank You, we’ve moved this point to discussion.

Point 6:. General comments

The manuscript was not well written, with typo and unstructured sentences.  Eg the 4th point in the conclusion.

Response 6: Yes, we totally agree. We’ve performed the Proofreading, and the English in paper is now improved.

Round 2

Reviewer 1 Report

none

Author Response

Dear eviewer, thank You for reviewing our manuscript.

Additional improvements are made: repetitive sentences and tabled are removed and additional spell check is performed.

Reviewer 2 Report

The revised manuscript is much improved.  

minor comment:  Table 3 is repetitive. Perhaps can merge with Table 4.

Author Response

Dear reviewer, thank You for reviewing our manuscript.

We've completed the imrovements: Table 3 and Table 4 are merged and repetitive sentences are removed.